

# Minimal influence of future Arctic sea ice loss on North Atlantic jet stream morphology

Yvonne Anderson[1], Jacob Perez[1], Amanda C. Maycock[1]

[1]Institute for Climate and Atmospheric Science, School of Earth and Environment, University of Leeds, Leeds, UK

*Correspondence to*: Yvonne Anderson (ee22ya@leeds.ac.uk)

**Abstract.** The response of the North Atlantic jet stream to Arctic sea ice loss has been a topic of substantial scientific debate. Some studies link declining Arctic sea ice to a weaker, wavier jet stream, which potentially increases the occurrence of extreme weather events. Other studies suggest no causal link between Arctic sea ice loss and the jet stream, instead attributing jet variations to internal variability. Current methods for characterising
the low-level jet typically use zonal wind speeds averaged over the North Atlantic sector, which can result in the loss of important aspects of jet morphology. This study uses a new 2-dimensional feature-based method to investigate the winter low-level jet response to future Arctic sea ice loss using idealised prescribed sea ice experiments from the Polar Amplification Model Intercomparison Project (PAMIP). In contrast to earlier studies that have focused on seasonal average changes, this study also explores how daily jet variability is altered by sea
ice loss. The results show a significant equatorward shift in mean jet latitude for three of the six PAMIP models analysed, with a multi-model mean jet shift of -0.8 ± 0.1°. However, there is no change in jet speed and jet tilt across all models and no robust change in jet mass (area-weighted speed). Three of the six models show an increase in the frequency of split jet days, but this does not strongly affect the overall distributions of daily jet latitude, speed and mass. Likewise, the results show no significant change in the daily variability of jet features and changes in
interannual variability are inconsistent between the models. The results extend previous studies characterising jet response from a zonally averaged perspective, and suggest it is unlikely that future Arctic sea ice loss will cause significant weakening of the North Atlantic jet stream or an increase in jet variability.

## 1 Introduction

In recent decades the Arctic has warmed at an accelerated rate compared to the global average, in a process known
as Arctic Amplification (Serreze and Francis, 2006; Serreze et al., 2009; England et al., 2021; Rantanen et al., 2022). As the Arctic has warmed, there has also been a rise in the frequency of extreme weather events in northern mid-latitudes, causing substantial interest in the possible role of Arctic warming in driving midlatitude extremes (Francis and Vavrus, 2012; Cohen et al., 2014; Barnes and Screen, 2015; Shepherd, 2016; Cohen et al., 2020).

One possible mechanism linking Arctic Amplification and extreme weather is through impacts on the midlatitude
eddy-driven jet stream. Arctic Amplification is partly driven by sea ice loss, which weakens the meridional temperature gradient in the lower troposphere, potentially weakening the jet stream and causing an equatorward jet shift (Barnes and Screen, 2015; Cohen et al., 2020; Petoukhov and Semenov, 2010; Liu et al., 2012; Francis and



Vavrus, 2012, 2015). This weakening may lead to increased wave amplitude or 'jet waviness', slower Rossby wave phase speeds, and more persistent weather patterns (Screen and Simmonds, 2014). Through these mechanisms,

several studies have linked Arctic sea ice loss and recent mid-latitude weather, for example to regional cooling trends over North-East America and Eastern Eurasia (Outten and Esau, 2012; Inoue et al., 2012; Tang et al., 2013; Kug et al., 2015; Mori et al., 2019).

Other studies question the link between Arctic sea ice loss and extreme weather, with some model studies showing little or no changes in extremes with sea ice loss, and instead attributing observed events to internal climate

variability (McCusker et al., 2016; Blackport et al., 2019; Koenigk et al., 2019; Cohen et al., 2020). Smith et al. (2022) show a robust winter average weakening and equatorward zonal mean jet shift due to imposed Arctic sea ice loss in a multi-model ensemble. However, the amplitude of the signal was small compared to interannual variability, indicating only a weak atmospheric response to sea ice loss. The argument of Arctic warming driving a wavier jet stream has also been disputed, with some studies showing no decrease in wave speeds or an increase

in wave extents (Barnes, 2013; Hassanzadeh et al., 2014). Moreover, while the observational record spanning 1979 to 2012 suggests a correlation between Arctic sea ice loss and winter Eurasian cooling trends, extending the record to present day reveals a diminishing relationship (Blackport et al., 2019; Blackport and Screen, 2021; Smith et al., 2022). Therefore, the extent to which Arctic sea ice loss influences the jet stream and regional midlatitude climate is not fully understood (Box 10.1. in Doblas-Reyes and Sörensson, 2021).

This study focuses on the lower tropospheric component of the North Atlantic jet stream. A widely adopted framework for characterising the low-level jet is the Jet Latitude Index (JLI; Woollings et al., 2010). The JLI has been applied to simulations from the Polar Amplification Model Intercomparison Project (PAMIP; Smith et al., 2019), which shows a small equatorward jet shift in the winter mean, but no robust change in jet speed across models (Ye et al., 2023). However, Ye et al. (2023) focused on seasonal mean changes and neglected short-term

jet variability, which is often associated with extreme weather events. Furthermore, the 1-dimensional view of the jet structure used by the JLI (Woollings et al., 2010) neglects important jet characteristics related to tilted, split, weak and broad jets (Perez et al., 2024, in press). Here, we use a new feature-based jet identification method (Perez et al., 2024, in press) applied to PAMIP experiments to explore the effect of future Arctic sea ice loss on North Atlantic jet structure and its variability.

The remainder of the paper is laid out as follows: Section 2 describes the datasets and methodology used for characterising the jet stream. Section 3 presents an evaluation of the PAMIP models' representation of the present day jet, followed by an assessment of the impact of future sea ice loss on the jet with a focus on daily and interannual variability. We also quantify the frequency of split jets and discuss their importance to the changes in jet morphology. Finally, in Section 4 we discuss the limitations of the study and summarise our conclusions.



## 2 Datasets and methods

### 2.1 PAMIP model experiments

PAMIP aims to improve understanding of the processes driving polar amplification and the consequences for the climate system, with a key goal of constraining the atmospheric response to Arctic sea ice loss (Smith et al., 2019). This study uses two large ensemble atmosphere-only experiments from PAMIP. PAMIP experiment 1.1 simulates present-day climate forced by present-day sea surface temperatures (SSTs) and Arctic sea ice concentration (SIC). PAMIP experiment 1.6 is forced by present-day SSTs and projected future Arctic SIC under a 2°C global warming scenario relative to pre-industrial climate. Present-day conditions are constructed from the monthly mean climatology between 1979 and 2008 from the Hadley Centre Sea Ice and Sea Surface Temperature observational dataset (HadISST; Rayner et al., 2003). Future conditions are obtained from Representative Concentration Pathway 8.5 (RCP8.5) simulations from phase 5 of the Coupled Model Intercomparison Project (CMIP5; Taylor et al., 2012). Taking the ensemble mean SIC for CMIP5 simulations results in a poor representation of the ice edge. To address this issue, future SIC projections are constrained by present-day observations to ensure a more accurate representation. Additionally, in regions where the difference between present-day and future SIC is greater than 10%, present-day SSTs are replaced by future SSTs in experiment 1.6.

PAMIP experiment 1.1 and 1.6 are time-slice experiments, with initial conditions taken from historical Atmospheric Model Intercomparison Project (AMIP) simulations starting on 1st of April 2000 (Taylor et al., 2000). Each ensemble member runs for 12-14 months, with the first two months discarded for model spin-up. The selected PAMIP models (Table 1) are those that provide daily zonal wind speeds, which allow characterisation of the eddy-driven jet stream on daily timescales. Data from PAMIP models was obtained from the Earth System Grid Federation website (CEDA, 2023). Models provide at least 100 members and have been re-gridded to a common grid of 2.81° x 2.81°, which is the resolution of the coarsest model analysed (CanESM5). The focus of the analysis in this study is on the North Atlantic region during winter.

**Table 1:** Selected PAMIP models used in this study, their ensemble size and horizontal resolution.

| Model | Ensemble size | Horizontal resolution (° lat x ° lon) |
|---|---|---|
| AWI-CM-1-1-MR (Semmler et al., 2019) | 100 | 0.55 x 0.83 |
| CanESM5 (Sigmond et al., 2019) | 100 | 2.81 x 2.81 |
| FGOALS-f3-L (He and Bao, 2019) | 100 | 0.55 x 0.83 |
| HadGEM3-GC31-MM (Eade, 2020) | 300 | 0.55 x 0.83 |
| IPSL-CM6A-LR (Boucher et al., 2019) | 200 | 1.26 x 2.50 |



| MIROC6 (Mori, 2019) | 100 | 1.41 x 1.41 |
|---|---|---|

The largest reductions in sea ice between the present-day and future experiment are over Hudson Bay, the Sea of Okhotsk and the Barents and Bering Seas (Figure 1a), resulting in the largest near-surface air temperature anomalies in these regions (Figure 1b).

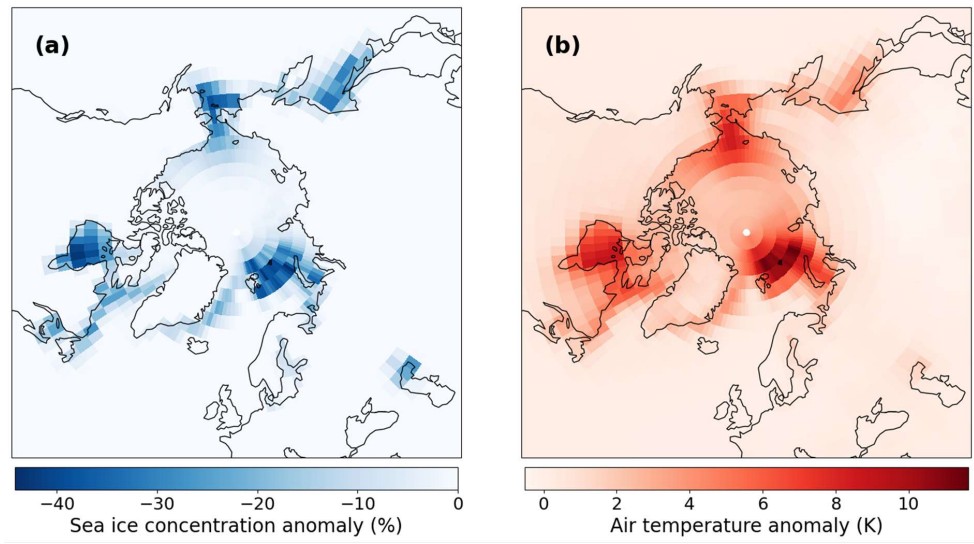

**Figure 1:** DJF multi-model mean difference in (a) SIC and (b) near-surface air temperature between present-day and future
PAMIP experiments.

## 2.2 Observation based datasets

Daily zonal wind speed data from the ERA5 reanalysis (Hersbach et al., 2020) is used to assess the performance of the PAMIP models. ERA5 was re-gridded to a common 2.81° x 2.81° resolution for consistency with the PAMIP models. Jet features were calculated for all winters over the period 1979-2020.

## 2.3 Feature based jet identification

This study applies a new feature based approach for diagnosing the low-level North Atlantic jet stream (Perez et al., in press), with the aim of characterising the association of the jet structure with sea ice loss in more detail than previous studies. A common first step for diagnosing the eddy-driven jet stream is to take the average zonal wind speed across a longitudinal sector. The method applied in this work starts from the non-averaged zonal wind field
at 850 hPa ($U_{850}$) (Woollings et al., 2010; Madonna et al., 2017). The wind field is constrained to the North Atlantic



sector (0-60°W, 15-75°N) and the winter season (December, January, February (DJF)). A 10-day low-pass Lanczos filter with a window of 61 days is applied to remove short-timescale fluctuations and for closer comparison with previous methods (Woollings et al., 2010), though this choice does not significantly alter the daily jet statistics (Perez et al., in press).

The feature-based approach identifies westerly jets by using a minimum zonal wind threshold of 8 ms$^{-1}$. To capture large-scale, zonally oriented jets, a minimum geodesic jet length of 1661 km and a minimum longitudinal extent of 20° are also applied. The latter two thresholds ensure the jet objects identified are zonally orientated and also allows for days with no well defined jet to be identified. Firstly, the grid point with the maximum $U_{850}$ above the 8 ms$^{-1}$ threshold is located. Secondly, all surrounding grid points with $U_{850}$ greater than 8 ms$^{-1}$ are connected to create

an outline of the jet object (Figure 2). The process for identifying a jet object can then be repeated for the next largest $U_{850}$ above the threshold, allowing multiple jet objects to be identified on one day.



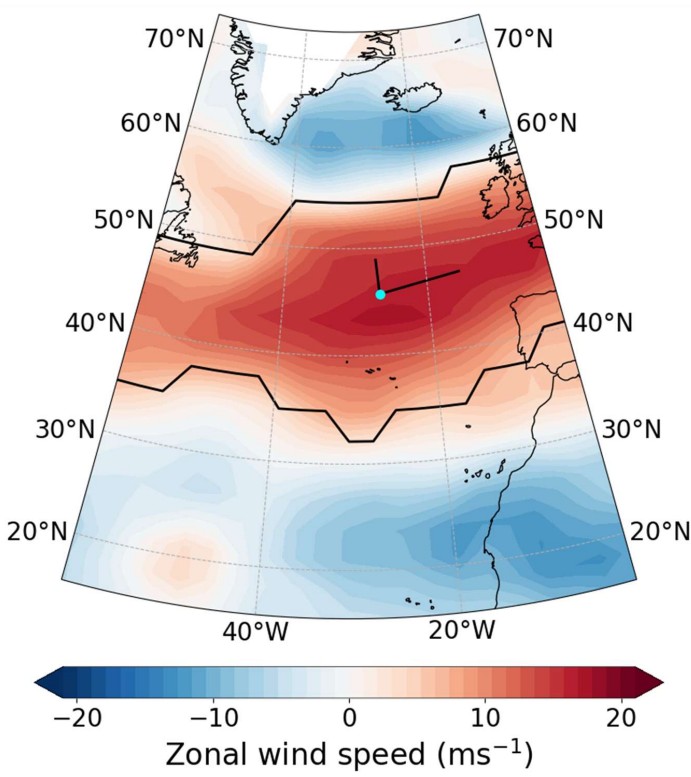

**Figure 2:** Application of the jet identification method to U$_{850}$ in the North Atlantic region on an example day. The black contour shows the outline of the jet object found on this day, the blue dot shows the centre of mass of the object giving the jet position, with the minor and major axes denoted by the black lines.

### 2.3.1 Spatial moment analysis

Once a jet object has been identified, morphological jet features are determined using spatial moment analysis. Moments and centralised moments of the U$_{850}$ ($\lambda$,$\phi$) field are calculated over a 2-dimensional object (R) using equation 1 and 2 respectively:

$$M_{pq} \; = \; \iint_R \lambda^p \phi^q \, U_{850}(\lambda, \phi) dA \tag{1}$$

$$\tilde{M}_{pq} \; = \; \iint_R (\lambda - \bar{\lambda})^p \, (\phi - \bar{\phi})^q \, U_{850}(\lambda, \phi) dA \tag{2}$$

where p is the order of the moment in the longitudinal direction and q the order in the latitudinal direction.





The latitude of the jet centre of mass ($\bar{\phi}$) is calculated using equation 3:

$$\bar{\phi} = \frac{M_{01}}{M_{00}} \tag{3}$$

where $M_{00}$ is the jet mass. The Jet speed ($U_{mean}$) is the average wind across the jet object which is defined by equation 4:

$$U_{mean} = \frac{M_{00}}{\int_R dA} \tag{4}$$

Finally, jet tilt ($\alpha$) is the angle between the longitudinal axis and the major axis of the jet and is calculated using equation 5:

$$\alpha = \frac{1}{2}\tan^{-1}\left(\frac{2\tilde{M}_{11}}{\tilde{M}_{20} - \tilde{M}_{02}}\right) \tag{5}$$

2-dimensional moment analysis has been applied in previous studies to characterise the stratospheric polar vortex (Waugh, 1997), particularly to assess vortex variability and to distinguish between split and displacement events (e.g., Mitchell et al., 2013; Maycock and Hitchcock, 2015). Feature based methods for characterising the jet stream have been applied previously to reanalysis (Limbach et al., 2012; Spensberger and Spengler, 2020). However, those

studies focus on the upper-level jet structure rather than the lower tropospheric component which is the focus of this study.

## 2.4 Statistical methods

The initial analysis uses the jet latitude, speed, mass, tilt and area for the largest mass jet object on each day. The significance of the difference in sample means and cumulative distribution functions was assessed using a two-

sample Student's t-test and Kolmogorov-Smirnov (K-S) test at the 95% confidence level. Prior to computing the t-test and K-S-test p-values, the effective number of degrees of freedom in the data was determined where the effective degrees of freedom ($N_{eff}$) is defined as:

$$N_{eff} = N\frac{1-r}{1+r} \tag{6}$$

where N is the sample size and r is the lag-1 autocorrelation.

This process accounts for autocorrelation in the jet variables at daily timescales and results in a reduced number of independent data points relative to the total sample size. For models that show a significant difference in daily mean and cumulative distribution function (i.e. a significant p-value for the t-test and K-S test), the difference in mean was subtracted from the future distribution and the K-S statistics recalculated. This process allowed an evaluation of whether differences in the distribution could be explained by a change in the mean or whether higher

order moments such as variance and skewness also contribute to the difference.

We compare standard deviations between time periods to determine the effect of Arctic sea ice loss on the daily and interannual variability of jet features. We also calculate the skewness of the jet features to highlight any





asymmetry in the distributions. To determine whether the difference in standard deviations is significant, outputs from present-day and future simulations were bootstrapped with replacement and the difference in their standard deviation calculated. The difference in standard deviation was considered significant if it fell outside the 95th percentile of the bootstrapped differences. For interannual variability, the winter mean for each simulation was bootstrapped and the standard deviation of the resampled means was calculated. The difference between the present-day and future simulations was then taken to determine the range of interannual variability.

## 3. Results

### 3.1 Effect of Arctic sea ice loss on winter mean circulation

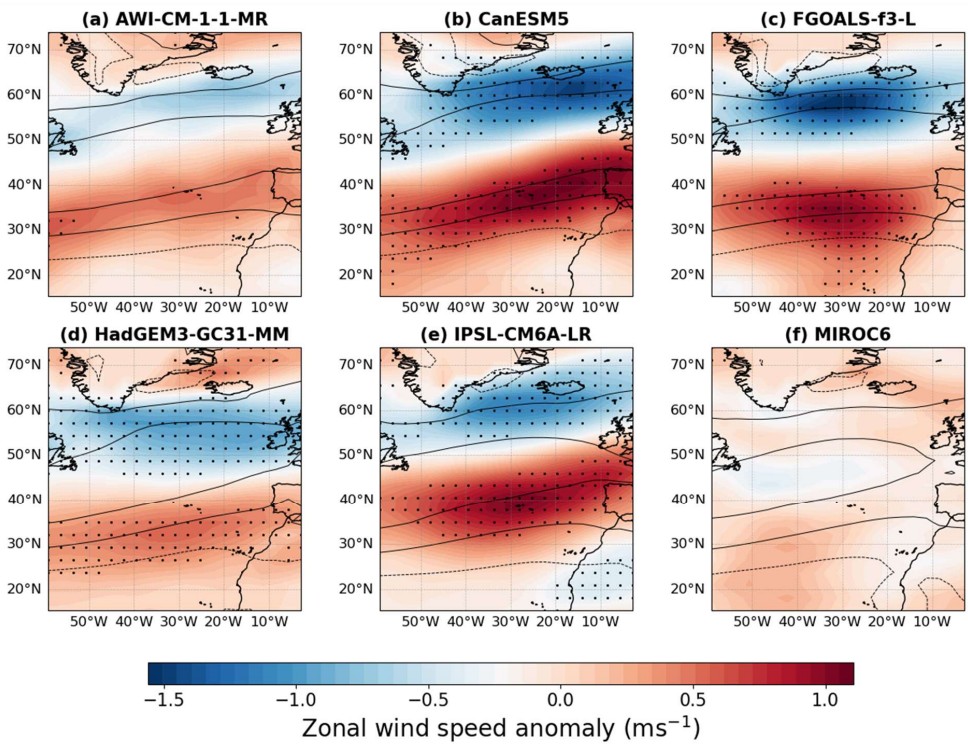

**Figure 3:** DJF ensemble mean $U_{850}$ anomaly in selected PAMIP models between simulations forced by present-day and future SIC. Stippling indicates objects where the difference is significant based on a two-sample Student's t-test at the 95% confidence level. Black contours signify the present-day winter $U_{850}$ climatology at 5 ms$^{-1}$ intervals with dashed lines showing negative values.



The winter mean North Atlantic $U_{850}$ anomaly due to future Arctic sea ice loss is shown in Figure 3. CanESM5, FGOALS-f3-L, HadGEM3-GC31-MM and IPSL-CM6A-LR show a decrease in wind speed at 50-60°N and an increase near 30-40°N (Figure 3 b-e), corresponding to an equatorward jet shift. AWI-CM-1-1-MR shows a similar response to HadGEM3-GC31-MM, but the difference is non-significant because of the smaller ensemble size (Table 1). MIROC6 shows a very weak and non-significant $U_{850}$ response, which is consistent with the weak JLI response in MIROC6 found by Ye et al. (2023).

Smith et al. (2022) found a mean equatorward jet shift due to Arctic sea ice loss in the PAMIP models based on the zonal mean zonal wind. Figure 4 shows a comparison between the zonal mean $U_{850}$ response analysed by Smith et al. (2022) and the North Atlantic average $U_{850}$ response. The zonal wind response index (ZWRI) is calculated as the difference in zonal wind anomaly between 30-39°N and 54-63°N (Smith et al., 2022). For five of the six PAMIP models analysed, the ZWRI for the North Atlantic sector is greater than for the zonal mean, with the response in five models being 0.5-1 ms$^{-1}$ (around a factor 2-3) larger. Thus, the overall weak ZWRI identified by Smith et al. (2022) is, on average, associated with a stronger local equatorward jet shift in the North Atlantic, with the exception of MIROC6 which shows a weak change in the North Atlantic than the zonal mean. This demonstrates it is important to examine the regional response to sea ice in individual basins.





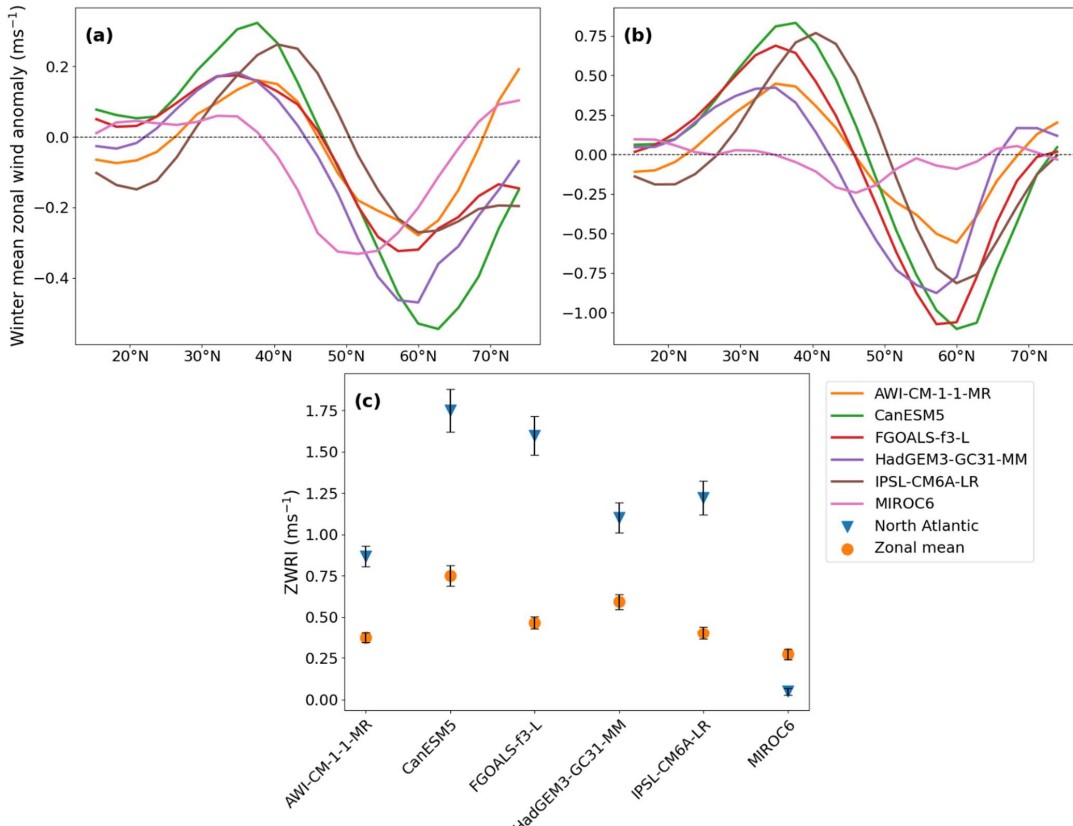

**Figure 4:** DJF mean $U_{850}$ anomaly due to Arctic sea ice loss in PAMIP models for a) zonal mean and b) North Atlantic (0-60°W) mean. Note the different vertical scales on (a) and (b). (c) the ZWRI of Smith et al. (2022) for the zonal mean (yellow circles) and North Atlantic sector (blue triangles).

### 3.2 Model performance for climatological jet structure

Before analysing the changes in jet characteristics under future sea ice loss, we first compare the present day SIC PAMIP experiments with ERA5 for daily jet latitude, speed and tilt (Figure 5). The comparison is not perfect because the PAMIP models do not include any year-to-year variation in boundary conditions that will contribute to variability in ERA5, but it gives an indication of their capability.



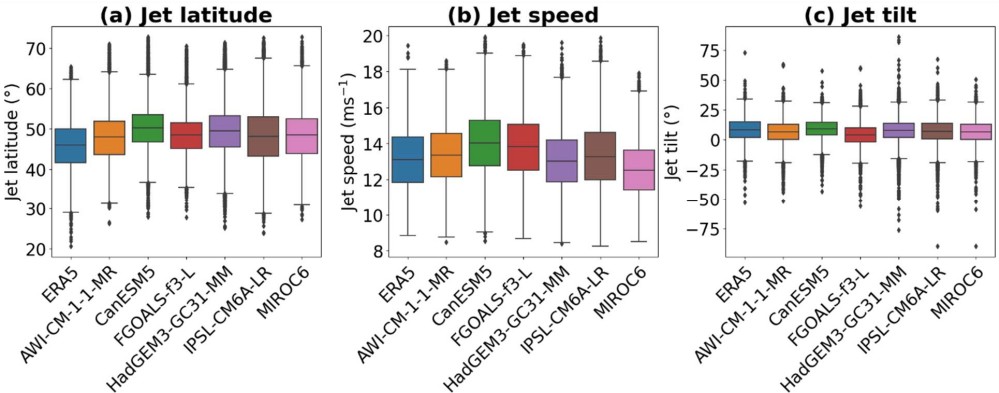

**Figure 5**: Distribution of daily jet features for present-day PAMIP models compared with ERA5 reanalysis. Comparisons shown are for a) jet latitude, b) jet speed and c) jet tilt. Coloured boxes show the interquartile range, the horizontal line shows the median, the whiskers show the overall distributions with diamonds representing outliers.

All PAMIP models show a climatological poleward bias in jet latitude, with a multi-model mean difference of 2.6°
compared to ERA5. A similar poleward bias in the North Atlantic jet has been previously shown for CMIP5 models using the JLI (Iqbal et al., 2018). ERA5 jet speed and jet tilt largely lie within the PAMIP model spread, with differences between the multi-model mean and reanalysis of 0.1 ms$^{-1}$ for jet speed and -1.3° for jet tilt. This analysis indicates that PAMIP models generally perform well for characterising the present day jet morphology, but the differences should be considered when interpreting model results.

### 3.3 Effect of Arctic sea ice loss on daily jet morphology

#### 3.3.1 Jet latitude ($\bar{\phi}$)

Distributions of jet latitude of the largest mass jet object on each winter day are shown in Figure 6. The jet shifts equatorward in the CanESM5, FGOALS-f3-L and HadGEM3-GC31-MM models between present-day and future simulations (Fig. 6 b-d), with a mean shift in these models of -0.8 ± 0.1°. Removing the mean difference from the future distribution results in a non-significant p-value for the K-S test, which shows that the overall differences in jet latitude distributions can be explained by a change in mean. For AWI-CM-1-1-MR, IPSL-CM6A-LR and MIROC6 models, the difference in mean jet latitude is non-significant (Fig. 6 a,e,f). Although IPSL-CM6A-LR shows a significant equatorward shift in the winter mean zonal wind speed response to Arctic sea ice loss (Fig. 4b), application of the feature-based daily jet identification method indicates a non-significant change to jet latitude.
AWI-CM-1-1-MR shows a similar mean decrease in jet latitude to HadGEM3-GC31-MM, but the difference is not significant due to the smaller ensemble size.



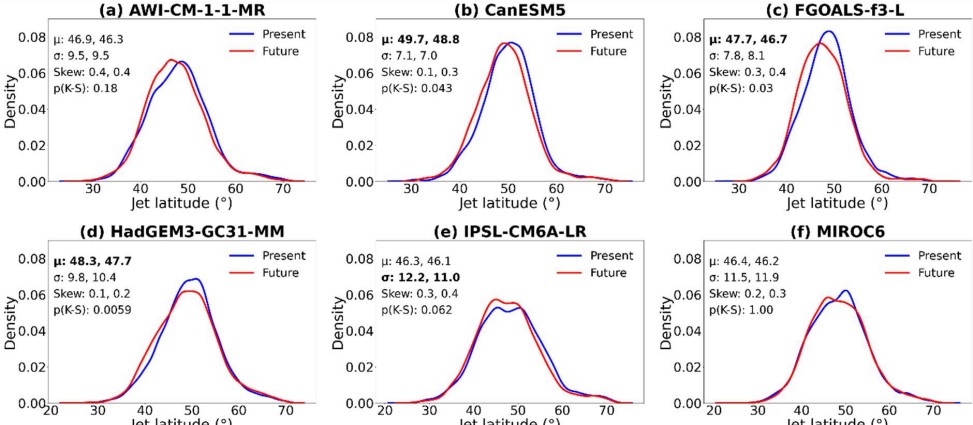

**Figure 6:** Distributions of daily jet latitude in winter for simulations forced by present-day (blue) and future (red) SIC. Data are for the largest mass jet object on each day and distributions have been fitted with a kernel density estimate. Ensemble mean (μ), standard deviation (σ), skew and K-S test p-value for the distributions are shown in the legend. Means are bold where the difference between present-day and future simulations is statistically significant based on a t-test at the 95% confidence level. Standard deviations are bold where the difference between time periods is greater than for random sampling.

Distributions of jet latitude in the present-day SIC experiment are positively skewed and generally become more positively skewed with future SIC. IPSL-CM6A-LR shows a small but significant reduction of -1.2° in daily jet latitude standard deviation between present and future, but for all other models the change in variability is non-significant. In contrast, four models show a significant change in interannual jet latitude variability. However, the sign of the response is not consistent across models, with AWI-CM-1-1-MR, and MIROC6 showing an increase, while CanESM5 and IPSL-CM6A-LR show a decrease. Key statistics for jet latitude in each model simulation are summarised in Table S1.

### 3.3.2 Jet speed ($U_{mean}$)

Distributions of jet speed for the largest mass jet object on each winter day are shown in Figure 7. The distributions show no significant change between present-day and future SIC experiment for all models, which contrasts previous studies that show a weakening of the jet with Arctic sea ice loss (Smith et al., 2022; Ye et al., 2023). It is interesting that while the winter mean, North Atlantic sector mean ZWRI shows a strengthening in several models (Figure 4c), this increase is not found when taking a view centred on daily jet objects. The daily variability of jet speed is not significantly different between simulations for all models, while four models do exhibit a notable shift in interannual variability. However, this difference is not consistent across models, with AWI-CM-1-1-MR and MIROC6 showing an increase, while FGOALS-f3-L and HadGEM3-GC31-MM show a decrease. It is also noted that the models indicating a significant alteration in jet speed interannual variability do not align with those showing significant changes in jet latitude interannual variability. Key statistics for jet speed are summarised in Table S2.





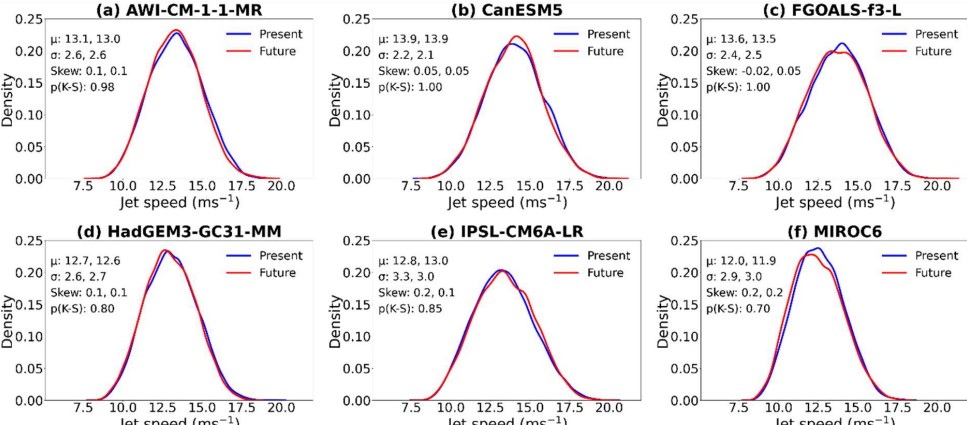

**Figure 7:** Distributions of daily jet speed in winter for simulations forced by present-day (blue) and future (red) SIC. Data are for the largest mass jet object on each day and distributions have been fitted with a kernel density estimate. Ensemble mean (μ), standard deviation (σ), skew and K-S test p-value for the distributions are shown in the legend.

### 245 3.3.1 Jet mass ($U_{mass}$) and tilt (α)

Distributions of jet mass for the largest mass jet objects on each winter day are shown in Figure 8. There is no significant difference in jet mass or its daily variability between present-day and future SIC experiments for five of the six models. The one exception is that HadGEM3-GC31-MM shows a significant difference in mean, but no difference in the distribution when assessed using a K-S test. The significant, but rather modest, difference in mean

250 may be caused by the HadGEM3-GC31-MM's larger ensemble size compared to the other models.





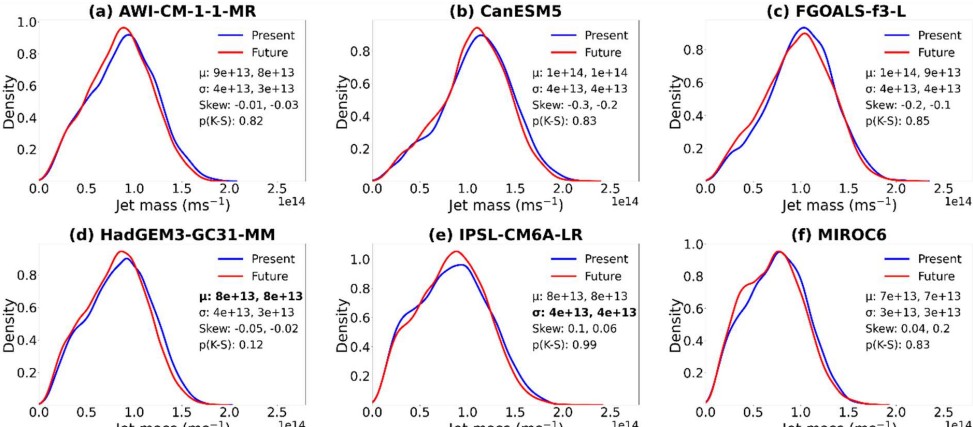

**Figure 8:** Distributions of daily jet mass in winter for simulations forced by present-day (blue) and future (red) SIC. Data are for the largest mass jet object on each day and distributions have been fitted with a kernel density estimate. Ensemble mean (μ), standard deviation (σ), skew and K-S test p-value for the distributions are shown in the legend. Means are bold where the difference between present-day and future simulations is statistically significant based on a t-test at the 95% confidence level. Standard deviations are bold where the difference between time periods is greater than for random sampling.

Three models show a significant change in jet mass interannual variability, but as for jet latitude and speed, the sign of the change is not consistent across models. Although the difference between simulations is not significant, for some models the shape of the distribution suggests a decrease in jet mass. Plotting the distribution of daily jet area (Fig. S1) shows that changes in the jet mass follow changes in area. This suggests that differences in the distribution of wind within the jet is not important for the differences in jet mass and instead the differences are dominated by a change in jet area. Key statistics are summarised in Table S3 for jet mass and Table S4 for jet area.

Finally, the jet identification method extracts the jet tilt. As with jet speed and jet mass, there is no significant change in daily jet tilt across models. Distributions of daily jet tilt are shown in Figure S2 and key statistics are summarised in Table S5. We find that the climatological mean daily jet tilt ranges from 4-9° across models, with no significant difference in daily or interannual variability under future Arctic sea ice forcing.

### 3.4 Occurrence of split jet and zero jet days

The occurrence of split jet and zero jet days in present-day and future simulations were quantified to further explore changes in jet morphology (Figure 9). Split jet days are defined as days when two jet objects are identified, while zero jet days are days when no jet is identified. We sum the occurrence of split jet days over all ensemble members and take the difference between simulations. To assess the significance of the difference, we generate bootstrapped samples of both simulations, calculate the occurrence of split jet days in each and take the difference. The spread of the difference in occurrence of split jet days between present-day and future simulations in bootstrapped samples





is shown in Figure 9a). A difference in split jet days is considered significant where the spread does not encompass
zero. The same process was followed to assess the number of zero jet days (Fig. 9b).

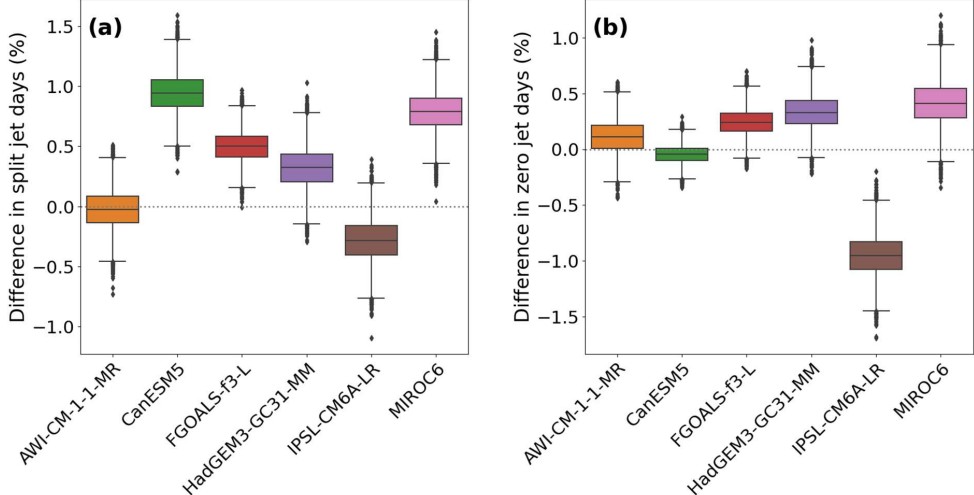

**Figure 9:** Difference in the occurrence of split jet and zero jet days between present-day and future simulations. The percentage
of a) split and b) zero jet days are shown between bootstrap resamples of present-day and future ensembles. Coloured boxes
show the first quartile, median and third quartiles of the differences, whiskers show the overall distributions and black
diamonds represent outliers.

The present-day multi-model mean percentage of split jet days is 3%. CanESM5, FGOALS-f3-L and MIROC6
models show a small but significant increase in the frequency of split jet days of around 0.5-1% between present-
day and future simulations. We also assess the significance in the change in frequency of split jet days by evaluating
the impact on jet latitude, as a significant increase in split jet days may result in broadening of the jet latitude
distribution, due to the increased instances of low and high latitude jets. There are no significant differences in
standard deviation between the simulations when the second jet object are included in distributions of daily jet
latitude (Fig. S3). While significant differences in split jet days are found in some models, the proportion of sample
days the change represents is small (<1.5%), which makes the impact on the overall distribution modest. The
occurrence of zero jet days shows no significant change between present-day and future for five of the six models.
However, IPSL-CM6A-LR shows a significant decrease in zero jet days.

## 4. Discussion

This study set out to characterise the effects of projected future Arctic sea ice loss on the winter North Atlantic jet
stream morphology. We have assessed how daily jet features are altered by sea ice loss by analysing daily zonal



wind speed data from present-day and future PAMIP simulations. Our approach is different from previous studies
that assess changes in the jet on seasonal timescales and use average zonal wind speeds across a longitudinal sector.
The analysis also covers changes in jet morphological features that have not been considered previously, including
jet tilt and area.

The equatorward jet shift shown by some models is consistent with the winter zonal mean perspective (Smith et
al., 2022). Likewise, the mean shift in daily jet latitude was -0.8 ± 0.09°, which is comparable to the multi-model
winter mean change of -0.6° found by application of the JLI to a larger set of PAMIP models (Ye et al., 2023).
Three models show a significant change in daily mean jet latitude (Fig. 6), compared with four models showing a
significant change based on the winter mean zonal wind speed (Figure 3). The non-significant response for daily
jet latitude in IPSL-CM6A-LR may be due to larger daily variability in the jet than seasonal variability. IPSL-
CM6A-LR also shows a decrease in the occurrence of zero jet days (Fig. 9b) captured by the jet identification
method. Removing zero jet days results in a non-significant difference in standard deviation between the
simulations compared to the range of standard deviations from random sampling. This suggests the significant
decrease in jet latitude daily variability for IPSL-CM6A-LR is due to the decrease in zero jet days.

The feature-based jet identification method allows for the quantification of split jet days, a detail that is missing
from previous studies. Three of the six PAMIP models show an increase in split jet days with future Arctic sea ice
loss. However, the increase is small (<1.5%) and the distributions of daily jet latitude are not affected. Split jets are
often characterised by one high latitude and one low latitude jet object, which means if the largest jet object was
the high latitude jet, it would positively skew the distribution. Likewise, if the increase in split jet days between
present day and future periods is significant, we expect to see broadening of the distribution in future. We find that
including the second largest jet objectwhen plotting the daily jet latitude distribution has minimal effect on the
difference in daily latitude or skew between simulations (Fig. S3). This result suggests the increase in split jet days
is not large enough to have a significant impact on the distribution of daily latitude with future sea ice loss relative
to the present-day.

The results suggest that Arctic sea ice loss has no effect on North Atlantic jet speed, which contradicts the argument
that Arctic sea ice loss will drive a weaker jet stream in the future (e.g Outten and Esau, 2012; Francis and Vavrus,
2015). Furthermore, this result does not align with the zonal mean perspective, which indicates a robust weakening
of westerly winds across all PAMIP models (Smith et al., 2022). Minimal influence on jet speed across all models
also contrasts with the JLI approach, which shows alterations in jet speed for HadGEM3-GC31-MM, IPSL-CM6A-
LR, and MIROC6, albeit with some models showing strengthening of the jet and others showing a weakening (Ye
et al., 2023). This analysis highlights that the result for jet speed is dependent on the approach for characterising
the jet.

Following changes in daily jet speed, the results show no significant change in jet mass in response to Arctic sea
ice loss. As jet mass is the area-weighted jet speed, it is expected that models exhibiting an equatorward shift in jet
latitude and no change in jet speed between simulations (CanESM5, FGOALS-f3-L and HadGEM3-GC31-MM),
increase in jet mass due to an increase in the area covered by the jet, owing to the Earth's curvature. However, the
models that show a significant equatorward shift show no significant change in jet mass. The distributions show



that changes in the jet mass follow changes in area and although the difference is not significant, some models appear to show a slight decrease in jet mass and area with future sea ice loss (Fig. 8 and Fig. S1). The daily distributions only show the mass of the largest jet object, not accounting for days where the jet is split across multiple objects, which could explain the apparent decrease in area covered by the jet and thus mass. However, the

models that show an increase in split jet days do not significantly decrease in jet mass in future simulations. The lack of influence of split jet days also aligns with there being no significant difference in jet speed between simulations as split jets are associated with weakening of zonal wind speeds. Similarly, while IPSL-CM6A-LR shows a significant decrease in the number of days where zero jet objects are identified (Fig. 9b), the decrease is not large enough to significantly affect the distribution of jet speeds in the future simulation.

Jet tilt is an aspect that has been neglected in the zonal mean perspective of previous studies. However, the results show no significant changes in jet tilt between present day and future simulations. A wavier jet may be expected to occur alongside higher amplitude variations in jet tilt, but this is not seen, which suggests the jet waviness theory (Petoukhov et al., 2013; Francis and Vavrus, 2015) is not supported by PAMIP simulations.

The results also show consistent daily variability and inconsistent changes in interannual variability for all jet

features between present-day and future simulations. The modelled differences in jet features are all smaller than the interannual variability, which is consistent with previous studies focusing latitude and speed (Smith et al., 2022; Ye et al., 2023). The non-significant change in jet daily variability does not support the idea of the jet stream becoming wavier with future Arctic sea ice loss and aligns with the non-significant effect on jet speed, as jet waviness is associated with a weaker jet. As extreme weather is associated with a high amplitude, wavy jet stream

and changes on synoptic timescales, the results suggest it is unlikely that changes in the jet due to Arctic sea ice loss will cause more frequent extreme weather events in the future.

The analysis in this study is limited to an atmosphere-only diagnosis of the jet response to Arctic sea ice loss. The jet response may also be influenced by atmosphere-ocean coupling (Deser et al., 2015), but this is beyond the scope of this study. While there are some coupled atmosphere-ocean sea ice perturbation experiments in PAMIP, they

did not provide daily frequency output needed for this study. Furthermore, we were only able to use a subset of PAMIP models that provided daily zonal wind speed data for the atmosphere-only experiments. A further caveat is the potential role of the signal-to-noise problem, which affects simulations of the North Atlantic atmospheric response to Arctic sea ice loss (Smith et al., 2022). This effect may mean the modelled changes in the jet are an underestimation, despite the use of large ensembles in PAMIP simulations. Nevertheless, the analysis of high

frequency jet variability is important given the hypothesised links to extreme weather events. The model results presented here do not support a strong role for Arctic sea ice loss in driving increased jet variability and associated extreme events in boreal winter.

## 5. Conclusion

This study aimed to improve the understanding of the influence of Arctic sea ice loss on the North Atlantic jet

stream. We have analysed daily zonal wind speed data from the Polar Amplification Model Intercomparison Project





(PAMIP), which provides simulations of climate forced by present day and future sea ice concentrations. While previous studies have been limited to a zonally or sectorally averaged, 1-dimensional assessment of the jet (Smith et al., 2022; Ye et al., 2023), this analysis has given a more detailed view of the North Atlantic jet morphology by characterising additional jet features. As such, a new feature-based method, which does not rely on averaging over a longitudinal sector, was used to quantify changes in jet latitude, speed, tilt, mass and area. The analysis also goes beyond a study of seasonal average changes in the jet stream, by assessing the effect of Arctic sea ice loss on daily jet feature variability and the frequency of split jets.

Three of the six PAMIP models analysed indicate an equatorward shift in mean jet latitude with future Arctic sea ice loss, with a multi-model mean shift of -0.8 ± 0.09 °. This result generally aligns with the winter mean circulation response in the North Atlantic. One model shows a significant seasonal mean equatorward shift but insignificant shift in daily jet latitude between present-day and future simulations. This discrepancy is likely due to larger daily variability and the ability of the jet identification method to define zero jet days. The equatorward jet shift is comparable to the zonal mean perspective of previous studies and the application of the JLI (Smith et al., 2022; Ye et al., 2023).

None of the PAMIP models show a significant change in jet speed, which contrasts with the zonally averaged seasonal mean perspective of previous studies (Smith et al., 2022; Ye et al., 2023), and suggests the jet stream will not weaken with projected future Arctic sea ice loss. Similarly, the models show no significant influence of sea ice loss on the jet mass, area and tilt, the latter of which suggests it is unlikely that the jet will become wavier in future. The feature-based jet identification method also allows for the quantification of split jet days, a detail that has not been assessed previously. Three of the six PAMIP models indicate a significant increase in the number of split jet days in future, but the increase is small (<1.5%) and does not significantly alter the morphology of the jet. There is no change in daily variability and no robust change in interannual variability for all jet features between present and future simulations. This result suggests that future Arctic sea ice loss is unlikely to be a driver of increased extreme weather events, through enhanced variations in the jet stream.

The study is limited to an atmosphere-only assessment of changes in the jet and uses selected PAMIP models that allow analysis of daily variability. Furthermore, the modelled changes in jet features may also be too weak due to the signal-to-noise problem (Scaife and Smith, 2018). The results presented for these models do not provide evidence of a significant influence of future Arctic sea ice loss on jet morphology in boreal winter. Instead, the results suggest that future Arctic sea ice loss will have minimal effect on jet stream features, jet variability and the associated extreme weather events.

## Code and data availability

Data from PAMIP models used in this study can be freely downloaded from the Centre for Environmental Data Analysis (CEDA) portal on the Earth System Grid Federation website (https://esgf-index1.ceda.ac.uk/search/cmip6-ceda/, CEDA, 2023). Jet feature data that are required to create the study figures are available to download from https://doi.org/10.5281/zenodo.8279707 (Anderson, 2023). ERA5 reanalysis data



used for the model evaluation are available to download from the Copernicus Climate Change Service (https://doi.org/10.24381/cds.143582cf, Hersbach et al., 2017). Code for the feature-based identification method is available at https://github.com/scjpleeds/EDJO-identification.

**Competing interests**

The author declares that they have no conflict of interest.

**Acknowledgments**

The author acknowledges the climate modelling groups for producing and making available the PAMIP model outputs analysed in this study. JP was funded by a studentship from the EPSRC Centre for Doctoral Training in Fluid Dynamics at the University of Leeds. ACM acknowledges funding from the EU H2020 CONSTRAIN project
and the Leverhulme Trust.

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
