# Peer review of "Minimal influence of future Arctic sea ice loss on North Atlantic jet stream morphology"

_EGUsphere, 2024_

## Author Comment (AC2)

**Reviewer Response 'Minimal influence of future Arctic sea ice loss on North Atlantic jet stream morphology' submitted to Weather and Climate Dynamics**

Yvonne Anderson, Jacob Perez, Amanda C. Maycock

October 2024

We thank the Editor for sourcing three detailed reviews of our manuscript. We are grateful to the reviewers for their time evaluating the manuscript and for their constructive suggestions for improvements. We have taken on board the comments and revised the manuscript accordingly. We respond to individual points below. The reviewer comments are in black, and the author responses are in blue.

**Kristian Strommen**

**General comments**

This article applies the recently developed jet tracking method of Perez et al. to the PAMIP simulations, in order to assess how Arctic sea-ice loss will impact daily and seasonal timescale jet morphology in boreal winter. The basic message of the paper is that there is basically no impact.

The paper is mostly clearly written, well structured and easy to read. It is quite a quick and easy paper to read, owing to the fact that it's a "null result" paper, which means there just isn't that much to say. I think the paper adds a useful contribution to the Arctic-ice-loss literature by examining more nuanced jet morphology on daily timescales. The null-result adds more weight to the idea that ice loss won't notably affect the jet, by showing that important changes aren't being concealed by large-scale zonal means that become visible in things like jet tilt etc (with the usual signal-to-noise paradox caveat applying).

I have two major comments and then a few minor comments. I hope the authors can address these without undue burden.

We thank Dr Strommen for his time in carefully reading the manuscript and for his supportive comments about the merits of our study.

**Major comments**

**Comment A** L180-185: I'm very confused about the two ZWRI indices being compared, and why they're not the same. Maybe I am being dense, but the authors don't provide enough information in the text

to be completely sure what exactly is being computed. My interpretation was that the "Smith et al version" was to take zonal averages (i.e., average across longitudes 0-60W), then latitudinal averages over the two latitude regions (54-63N and 30-39N), and then taking the difference of the two regions. My interpretation of your "North Atlantic average version" was that you first take latitudinal averages across those two regions, look at the difference at each longitude 0-60W, and then take the average of these differences. However, these two versions give the same number at the end, because everything is linear; the order in which latitudes and longitudes have been averaged has simply been switched. So this can't be a correct interpretation, as you show that the two ZWRI versions are different. Can you please clarify exactly what is being computed here and why the two versions differ exactly? Please clarify the text accordingly.

**Response**: The Smith et al. ZWRI is based on the zonal mean zonal wind not the North Atlantic basin only. This was noted in L177-178 of the manuscript though not explicitly linked to the calculation of the ZWRI. We therefore compare their zonal mean ZWRI with the North Atlantic ZWRI averaged over 0-60W. Smith et al. calculate their ZWRI based on the average wind between 150-600 hPa where the model zonal wind responses peak. In contrast, we calculate both indices at 850 hPa so they are directly comparable (i.e. any differences come from including regions outside the North Atlantic in the zonal mean rather than from the use of different vertical levels). We have clarified this in the revised manuscript as follows:

'Smith et al. (2022) found a winter mean equatorward jet shift due to Arctic sea ice loss in the PAMIP models based on the zonal mean zonal wind. However, changes in zonal wind in other regions may not reflect the local North Atlantic eddy-driven jet response. Therefore, in Figure 4 we compare the zonal mean U850 differences (Fig 4a) and the North Atlantic sector (0-60W) U850 differences (Fig. 4b). In both cases a dipole pattern is evident, with positive U850 at lower latitudes and negative U850 differences at higher latitudes, corresponding to an equatorward jet shift. However, for all models except MIROC6, the North Atlantic sector U850 differences are larger than the zonal mean U850 differences. This can be further seen by examining the zonal wind response index (ZWRI) from Smith et al. (2022) calculated as the difference in vertically-averaged (600-150 hPa) zonal mean zonal wind between two latitude bands (30-39°N and 54-63°N). We recalculate this for the North Atlantic sector only (0-60W) (Fig. 4c). Note in contrast to Smith et al. (2022) we use U850 for both calculations for consistency.'

**Comment B** L344-351: I feel some extra care is required with this paragraph.
Firstly, your measure of variability is just the standard deviation. Observing no change in the standard deviation does not mean that important things haven't changed about the jet variability, because there can be compensation between the types of jet variability. For example, there could be changes to the persistence of certain jet configurations. If the jet overall spans the same latitudes as it did before ice loss, but the persistence of certain jet configurations have increased at the expense of others, then one can easily end up observing no change in standard deviation. However, it is clearly of interest to extremes to know if some jet configurations have become more persistent or not. It has in fact been shown that CMIP6 models project a decrease in the persistence of certain jet configurations (https://doi.org/10.1029/2022GL100811), so this point isn't just academic. Your framework doesn't easily allow you to assess such changes, because it would require you to identify specific configurations of interest, i.e., "weather regimes" of some sort. I note that the ability to assess changes to persistence and occurrence separately is a nice

advantage of a "regime" approach, and can make sense to do even if you don't believe the specific regimes are intrinsic to the atmosphere. I don't expect the authors to compute persistence statistics, but this caveat should be explained.

Secondly, the authors assert that extreme weather is associated with a high-amplitude wavy jet stream. This is only true if you have decided up front what extreme weather you care about! A very strong zonal jet pointing right at the UK during winter can often cause flooding, because all the storms dump their rain on the UK. This is not a wavy jet stream, but it is certainly extreme weather! There is a general theme in a lot of literature on Arctic-ice-loss-may-or-may-not-cause-extreme-weather to never actually specify which extreme weather events one cares about and just assume it's all wavy. This may or may not be fine when restricting to summer (I don't know), but for winter it seems like an implicitly biased definition of "extreme weather". The authors should clarify the text to emphasise that you are only talking about a specific subset of extreme European winter weather.

Thirdly, you point out an increase in split jet days in 3/6 models. However, you argue that this change doesn't matter because (a) the number of split days is small to begin with and (b) the change is too small to affect the overall distribution of jet latitude etc. But if one is interested in extreme weather, then this dismissal doesn't seem reasonable. Extremes are also only a small percentage of all events, but we still care about them. Changes in frequencies of extremes will also necessarily be small, but again, we care about them. Furthermore, in summer, split jets have been linked to certain kinds of extremes (e.g. https://doi.org/10.1038/s41467-022-31432-y). Have the authors looked at all at what kind of winter extremes could be associated with split jet days? Or does there exist prior literature on this? If not, then it seems there is an important caveat to note here.

**Response**:

- The authors agree that the standard deviation of jet features is only one measure of jet variability and this caveat has been added to the manuscript.

- We have removed the discussion of extreme events and wavy jets as it's not central to our study. The revised text now reads: 'The total variability as measured by the standard deviation shows no significant changes in daily variability and inconsistent changes in interannual variability for all jet features between present-day and future simulations. The modelled differences in jet features are all smaller than the present day interannual variability, which is consistent with previous studies focusing latitude and speed (Smith et al., 2022; Ye et al., 2023). The rather weak/missing changes in standard deviation of jet parameters does not rule out the possibility for other changes in North Atlantic circulation that are not detected by this measure. For example, there could be changes in the frequency and/or persistence of certain weather regimes which may have compensating effects when viewed through the standard deviation. In future work, we will examine the relationship between the jet parameters and weather regime frameworks (e.g., Madonna et al., 2017).'

- We have updated the wording around the importance of split jets in the analysis. We note that while split jet days make up a small percentage of days and do not significantly affect the standard deviation of jet diagnostics, it does not mean they are not important. We also note

that previous analysis of jet splitting has focused on their influence on summer temperature extremes, therefore analysis of jet splitting using this identification method across seasons could be potential future work. We have also added a supplementary table detailing the percentage occurrence of days with zero, one and two identified jets (Table S6). L308-317 from the original manuscript describing the split jet days now reads: 'The feature-based jet identification method allows for the quantification of split jet days, a detail that is missing from previous studies. Three of the six PAMIP models show an increase in split jet days with future Arctic sea ice loss. However, the increase is relatively small (<1.5%) meaning the total standard deviation of daily jet latitude is not affected. Split jets are often characterised by one high latitude and one low latitude jet object. Therefore, a larger increase in split jet days between present and future could lead to a broadening of the jet latitude distribution. We find that including the second largest jet object to capture the split jets has a minimal effect on the difference in the standard deviation and skewness of daily jet latitude between the present and future sea ice simulations (Fig. S3). Although split jet days make up a small percentage of days overall, it is important to quantify them because they can be associated with blocked flow, resulting in more persistent and sometimes extreme winter weather in Europe. Further work could analyse jet splitting in different seasons using this jet identification method given the known links with summer heatwaves (Rousi et al., 2022).'

**Minor Comments**

**L98:** ERA5 has been regridded to 2.81 degrees. What effect does this have on the diagnosed jet variability compared to ERA5 at 1 degrees? I am unsure how much of the trimodal JLI pdf is maintained when computing JLI using 2.81 degree zonal winds. The troughs in the pdf are order 4-5 degrees across when computed using 1 degree winds, so it's easy to imagine a lot of this structure vanishing when using 2.81 degree winds. I would guess the impact on the jet morphology method are small (since everything is unimodal there) but it would be good to check and comment, perhaps including a supplementary figure.

**Response:** We find that regridding to 2.81° x 2.81° has minimal influence on the daily jet diagnostics. This has been tested for the AWI-CM-1-1-MR model and we show no significant changes in jet feature statistics when using the higher resolution wind data (0.55° lat x 0.83° lon). A supplementary figure (S4) has been added to address this comment and the following text at the end of Section 3.3: 'While the diagnostics were calculated with regridded U850 data to ensure consistency, we have tested the results using the native grid for the highest horizontal resolution model (AWI–CM-1-1-MR; 0.55° latitude x 0.83° longitude) and find that this does not significantly alter the results (Figure S4)'.

**Figure 3:** The use of the word "objects" is a bit strange. Maybe "gridpoints" would be better?

**Response:** The wording has been updated in the manuscript.

**Figure 4:** I strongly recommend giving (a) and (b) the same y-axis. It will make it easier to parse the key qualitative information for the reader.

**Response:** Thanks for this suggestion. The figure has been updated as suggested.

**L302-303:** About the IPSL model: is the p-value for change in jet latitude close to 0.05? If yes, that could indicate that the non-significance for jet latitude vs significance for U850 changes is partly just noise as well.

**Response:** For the IPSL model, the t-test p-value for change in jet latitude is 0.64 and the ks-test p-value is 0.062.

**L314:** There is a missing space.

**Response:** Space added in the manuscript.

**L353:** The paper https://doi.org/10.5194/wcd-3-951-2022 also argues that ice-ocean-atmosphere coupling may be important to simulate Arctic-midlatitude links and that such coupling may be missing in most models. I hope the authors will not consider it grossly inappropriate of me to suggest citing it, given its relevance here.

**Response:** We agree that this paper is relevant and have cited it in the manuscript.

**Raphael Köhler**

**General comments**

This study investigates the effect of future sea ice changes on the low-level jet. They therefore analyse 6 PAMIP models and quantify changes in the daily jet stream morphology using a new 2-dimensional feature-based approach based on Perez et al. (2024). This allows for not only quantifying jet strength and latitude but also jet tilt, split jet, and no jet days. Overall, the response of the jet stream to sea ice changes is small, with very few significant changes, which is mostly in accordance with earlier studies.

The paper is well written, the methods are sound and I enjoyed reading it. It is also a rather quick read, which is related to the fact that there are few significant or unexpected results. However, I still think it is a valuable contribution to further investigating the PAMIP experiments and the small influence of future Arctic sea ice loss on the jet stream within these experiments. I appreciate the use of daily data, but in my opinion, the potential of this approach was not fully realised (see major comment 2). I also have a concern about the methodology which is related to the coarse resolution (see major comment 1). Most of the minor comments are rather trivial. I hope that it is not too much trouble to address the major comments in a reply or the manuscript.

*We thank Dr Köhler for his time in carefully reading the manuscript and for his supportive comments about the merits of our study.*

**Major Comments**

Did you test how the regridding of all your model data onto the common coarse 2.81° x 2.81° grid affects the results? I fear that you lose quite a bit of information when you regrid to such a coarse resolution, in particular as the detected signals are rather small (e.g., 0.8 ± 0.1°). It would be nice to test this for one of the models with a higher resolution (or for ERA5). It might not play a large role in a climatological mean sense, but it would be good to test this.

**Response:** *We find that regridding to 2.81° x 2.81° has minimal influence on the daily jet diagnostics. This has been tested for the AWI-CM-1-1-MR model and we show no significant changes in jet feature statistics when using the higher resolution data (0.55 x 0.83 °). A supplementary figure (S4) has been added to address this comment and the following text at the end of Section 3.3: 'While the diagnostics were calculated with regridded U850 data to ensure consistency, we have tested the results using the native grid for the highest horizontal resolution model (AWI–CM-1-1-MR; 0.55° latitude x 0.83° longitude) and find that this does not significantly alter the results (Figure S4)'.*

Different studies have shown that timing is important when investigating the effects of AA / sea ice loss on circulation changes (e.g., Siew et al, 2020; Crasemann et al., 2017): Sea ice loss has been connected to a Scandinavian/Ural blocking type circulation anomalies in December / early winter and a NAO-pattern via a stratospheric pathway in February / late winter. Both circulation patterns are related to distinct changes in the North Atlantic jet stream. Although you use daily data your results are only shown as winter (DJF) mean. Did you investigate the changes in jet morphology on a monthly scale and do you see any differences when you do so?

**Response**:

- Thanks for this interesting comment. We have repeated the analysis for separate months to determine whether jet diagnostics vary within the winter season. Calculations of jet latitude, speed and tilt were repeated using jet objects found in December, January and February separately and compared to the winter mean. This information has been added into the manuscript at the end of Section 3.3.

- For the AWI-CM-1-1-MR model, there is a significant difference in mean jet latitude between present-day and future scenarios for December and January. There is also a significant difference in standard deviation between present-day and future in February. These differences are not observed for winter mean jet latitude response.

- For CanESM5, there is a significant decrease in jet tilt between present day and future scenarios in February. This is not observed in December, January or in the winter mean.

- For all remaining models there are no significant differences in individual winter months that are not observed in the ensemble mean jet latitude, speed and tilt.

- For CanESM5, FGOALS-f3-L and HadGEM3-GC31-MM models, there is a significant difference in mean jet latitude between present and future in winter, but this is not observed in December, January or February individually. Likewise for IPSL-CM6A-LR, we see a significant decrease in standard deviation between present-day and future scenarios in the winter mean, which is not observed in individual winter months. This result is likely due to the limited number of samples in individual months.

**Minor Comments**

L16: mean jet shift of -0.8 $\pm$ 0.1°, the negative sign is usually associated with a southward shift. However, it might make sense to actually write southward or equatorward" instead.

**Response:** The wording has been updated in the manuscript.

L30/31: Not sea ice but AA weakens the meridional temperature gradient. Might make sense to change the order there.

**Response:** The wording has been updated in the manuscript.

L61-62: I found this a bit confusing as Section 3 starts with the effect of Arctic sea ice loss on winter mean circulation

**Response:** The summary of the paper has been updated to include assessment on winter mean circulation in Section 3.

Fig. 3: The contours are somewhat unclear as they don't "close". Maybe it would help to add numbers?

**Response:** Numbers have been added to the contours in Figure 3.

Fig. 3: Maybe this figure could be skipped altogether as it is basically a reproduction of Fig. S1 from Ye et al. (2023), except for the AWI model.

**Response:** Although the analysis is similar to Ye et al. (2023), we believe it is important to include in the context of this study to have a direct comparison of the seasonal mean picture and daily variability.

L174: I agree that this is probably due to the smaller ensemble size, but you can't be 100

**Response:** The wording has been updated to account for other potential causes.

Fig. 4: I would suggest using the same scale for a) and b)

**Response:** The figure has been updated to give (a) and (b) the same y-axis.

Fig. 6,7,8: The text is somewhat small

**Response:** Font sizes of jet variable statistics in Figures 6-8 have been increased in the updated manuscript.

L237-238: I was confused about where to find the information on the interannual variability, as it's the only number not given within the figure. Nevertheless, it partly receives more attention than the other metrics. Hence, would it make sense to add this information to the figures?

**Response:** Standard deviations for interannual variability are shown in Tables S1-5.

L246-250: Isn't IPSL also an exception? At least the daily variation is given in bold font.

**Response:** Yes, IPSL is also an exception. This information has been added to the text.

Fig. 8: It might make sense to scale the ensemble mean and standard deviation of the jet mass by 1e14 as you do on the x-axis. At the moment the values are not helpful as one cannot identify differences.

**Response:** Mean and standard deviations in Figure 8 have been scaled to match the x axis in the updated manuscript.

L314: missing space between "object" and "when"

**Response:** Space added in the manuscript.

Discussion and Conclusion: I find the discussion and conclusion somewhat repetitive. Maybe merging them could help to reduce some of the repetition.

**Response:** The discussion and conclusion sections have been combined to avoid repetition.

L330-L339: I found these lines somewhat confusing. What is the main message of this paragraph?

**Response:** Thanks for pointing this out. We have shortened and restructured these lines to be clearer: 'Jet mass represents the area-weighted jet speed. Therefore, models that exhibit an equatorward jet shift and no change in jet speed (CanESM5, FGOALS-f3-L and HadGEM3-GC31-MM) might be expected to show an increase in jet mass due to an increase in the jet area owing to the Earth's curvature. However, none of the models show a significant change in mean jet mass. Some models appear to show a slight decrease in jet mass and area with future sea ice loss (Fig. 8 and Fig. S1), but the differences are not significant.'

Tables S1-S4: It would be nice to also have the information on significance, as given by bold font in the figures of the main text.

**Response:** Tables S1-S4 have been updated to include t-test p-values that inform ensemble means in bold in the main text.

**Russell Blackport**

**General comments**

This study investigates how Arctic sea ice loss impacts the North Atlantic jet stream using simulations from PAMIP. While previous studies have investigated changes in jet stream characteristics, this is typically done with zonal average wind speed and seasonal averages. This study examines the changes in jet morphology from daily data using a 2-dimensional feature-based method. The authors find that the jet shifts poleward in response to sea ice loss, but the jet speed, jet tilt, jet mass and daily variability of the jet features show little change. The authors conclude that future sea ice loss is unlikely to cause a significant weakening or increase in variability of the North Atlantic jet.

Overall, I thought this study was well done, and it could be an important contribution to the literature. The research questions are reasonable and well-motivated. The results are presented clearly (for the most part), and the conclusions are convincing. I appreciate the fact that the authors have chosen to publish and highlight the 'negative' results instead of searching to try to find 'positive' results to highlight (or not publish at all). I recommend publication after these minor comments are addressed.

We thank Dr Blackport for his time in carefully reading the manuscript and for his supportive comments about our study.

- One caveat that should be mentioned somewhere is that sea ice loss is just one potential driver of the midlatitude circulation and associated impacts. Just because there is no change in response to sea ice loss does not mean that there will be no change in response to global warming/increased CO2. It is likely that the authors understand this, but confusing the response to sea ice loss with the response to global warming is a common mistake/misunderstanding I see, so it might be good to be explicit about this.

**Response:** The authors acknowledge that Arctic sea ice loss is not the only driver of changes in midlatitude circulation. This caveat has been added to the Introduction: 'The eddy-driven jet stream plays a key role in regional weather and climate in the midlatitudes and can be affected by increases in greenhouse gases through several thermodynamic and dynamic mechanisms (Shaw, 2019).'

-L16: If I am understanding correctly (based on the description of this later at L207-209), the value for the multi-model mean shift here is the value for only the three models that show a statistically significant shift. This seems a bit misleading and somewhat cherry-picking. Isn't the more relevant value the multimodel mean over all models?

**Response:** Thanks for raising this point. The text has been updated to include the multi-model mean over all models instead of only the models that show a significant change in latitude.

-L45: Blackport and Screen (2020) could be cited here if the authors think it is useful. We showed little change in waviness in observed trends, trends in historical simulations, and in response to sea ice loss/Arctic amplification in targeted model experiments.

**Response:** The reference has been added to the updated manuscript.

-L47: Blackport et al. (2019) is not the most appropriate reference here. This study was primarily questioning the causality of the statistical relationship. Blackport and Screen (2021) was mostly about this as well, but we also did show some weakening of the relationship as well, so it is more appropriate.

**Response:** The reference has been removed.

-L50-59: A study that is missing in this discussion is Ye et al. (2024) who looked at daily variability of the North Atlantic jet in response to sea ice loss. They were still using the JLI index from Woollings et al. (2010), and they were only using one model, so the results presented here are still novel, but the study should be mentioned.

**Response:** Thanks for pointing out we had not cited this relevant paper. We have added a citation and some discussion of their results in the Introduction: 'Ye et al. (2024) examined daily jet variability using the JLI and found an equatorward shift of the jet and weakening westerly winds with Arctic sea ice loss; however, they only used one climate model so it is unclear if those findings reflect a wider range of models. Furthermore, the JLI used by Ye et al. (2024) adopts a 1-dimensional view of the jet structure (Woollings et al., 2010) which neglects important jet characteristics related to tilted, split, weak and broad jets (Perez et al., 2024).'

-L85: A caveat that should be mentioned is that 100 years may not be enough to separate the signal from internal variability in these experiments (e.g. Peings et al. 2021; Ye et al. 2024).

**Response:** This caveat has been added to the updated manuscript: 'We note that owing to weak signal-to-noise in the modelled response to Arctic sea ice loss (e.g Peings et al., 2021; Smith et al., 2022; Ye et al., 2024), 100 years of simulation may not be sufficient to isolate a forced signal.'

-L110-115: Could these thresholds cause any selection bias because you may not identify certain jet configuration that are less zonal/wavier? You do analyze the number of days where you can't identify the jet, but there could potentially be interesting things going on in these days that are missed.

**Response:** Perez et al., 2024 assess the robustness of the jet identification method to the thresholds used. The results show that jet latitude and tilt are not sensitive to changes in the zonal wind threshold between 6 and 11 m/s when applied to ERA5 wind data. Removing the jet length or longitudinal extent thresholds may increase the number of days when split jets are detected, as the jet regions will be smaller compared to a day with only one jet.

-L177-186: This analysis of ZWRI is a bit confusing. It should be clarified that the ZWRI from Smith et al. (2022) is calculated from the zonal mean over all longitudes. I am also not really sure what the point of this analysis is. Is it only to point out that the zonal wind response is stronger over the North Atlantic than in the zonal mean over all longitudes? The fact that the responses are stronger over the Atlantic (and Pacific) region (where the strongest jets are) has been shown in many studies, including Smith et al. (2022).

**Response:** We have clarified the calculation of the ZWRI calculation in the updated text. While other studies have shown the zonal wind responses within the storm tracks can be locally stronger than the zonal mean, we feel it is important to include this since our jet identification specifically focuses on the North Atlantic and we discuss changes in jet speed based on the algorithm, so this analysis of the Eulerian zonal winds sets the scene for the jet analysis.

-L209: Why only include the mean shift for the models that have a statistically significant jet shift? What is that value if you include all models?

**Response:** See response above. The multi-model mean equatorward shift is 0.6 +/- 0.1° and has been added to the text.

-L214 (and also L302, L375): Although it is not statistically significant, it does show a shift that is close to statistically significant (p=0.06), so it does not appear to be entirely inconsistent.

**Response:** This caveat has been added to the text in the results and discussion.

-L231-234 (also L320, L380): It is not clear to me that this contrasts with these previous studies. Ye et al. (2023) concluded that the jet speed response was weak, that models disagreed on the sign, and that there are only a few models that had a statistically significant response (and these did not agree on the sign). Overall, these seem consistent with the results found here.

**Response:** This section has been updated in the text. The results contrast as we do not see a significant response for jet speed, but consistent in that Ye et al. (2023) only show a weak response.

Smith et al. (2022) is interesting because they highlight the weakening of the midlatitude westerly winds (including in the abstract), but they never calculate any jet speed index. The figures themselves do not show clear evidence of the weakening of the jet because the weakening occurs only at higher latitudes and there is strengthening at lower latitudes (more indicative of an equatorward shift in the jet than a change in speed). The results of Smith et al. (2022) seem consistent with the results presented here even if some of the conclusions may not completely agree.
-L233-235: I do not understand this point. A positive ZWRI response as shown in Fig 4 is more indicative of an equatorward shift in the latitude which is seen when looking at the daily latitudes. It is not clear to me what the connection is between ZWRI and jet speed.

**Response:** Thanks for this important comment. We have calculated the jet speed based on the Eulerian seasonal mean zonal wind profiles in the North Atlantic. This analysis shows non-significant changes, except for HadGEM3-GC31-MM, which has a small jet weakening (-0.33 ± 0.25 m/s). This information has been added to Section 3.1.

-L288-290: What is the percentage of days ways with no identifiable jet in the present-day simulations?

**Response:** Across all models, the average percentage of days where no jet is identified with present-day sea ice forcing is 2.5%. A supplementary table has been added (Table S6) to give a summary of the occurrence of days where zero, one and two jet objects are identified across models.

-L299: Is this value for all models or only the three that have statistically significant response?

**Response:** See previous response, the shift in jet latitude has been updated to the multi-model mean.

-L349: Is it necessarily the case that a weaker jet is wavier? This is part of the Francis and Vavrus (2012) hypothesis, but I have not seen any theory or evidence that a weaker jet caused by Arctic warming would necessarily become wavier. A very recent study (Batelaan et al. 2024) finds a weaker jet in response to Arctic amplification in aquaplanet simulations, but a decrease in jet waviness.

**Response:** Thanks for this comment. This statement has been removed in the revised manuscript.

-L357: I don't think it has been conclusively proven that the signal-to-noise problem affects the response to sea ice loss and that this means the circulation response to sea ice loss is underestimated, although it is certainly plausible. This should be changed to 'which may affect...". This conclusion from Smith et al. (2022) seems to be somewhat challenged by the conclusions of Saffin et al. (2024).

**Response:** The text has been updated as suggested.

-There is a lot of repetition between the Discussion and Conclusion sections. These repetitions should be minimized and there should be a clearer distinction between Discussion and Conclusions sections. Another possibility is to combine them into one section.

**Response:** Thanks for this suggestion. The discussion and conclusion sections have been combined to avoid repetition.

**References**

Batelaan, T. J., C. Weijenborg, G. J. Steeneveld, C. C. van Heerwaarden, and V. A. Sinclair, 2024: The Influence of Large-Scale Spatial Warming on Jet Stream Extreme Waviness on an Aquaplanet. Geophysical Research Letters, 51, e2024GL108470, https://doi.org/10.1029/2024GL108470.

Blackport, R., and J. A. Screen, 2020: Insignificant effect of Arctic amplification on the amplitude of midlatitude atmospheric waves. Science Advances, 6, eaay2880, https://doi.org/10.1126/sciadv.aay2880.

Crasemann, B., Handorf, D., Jaiser, R., Dethloff, K., Nakamura, T., Ukita, J., Yamazaki, K. (2017). Can preferred atmospheric circulation patterns over the North-Atlantic-Eurasian region be associated with arctic sea ice loss?. Polar Science, 14, 9-20. https://doi.org/10.1016/j.polar.2017.09.002.

Madonna, E., Li, C., Grams, C. M., and Woollings, T.: The link between eddy-driven jet variability and weather regimes in the North Atlantic-European sector, Q. J. R. Meteorol. Soc., 143, 2960–2972, https://doi.org/10.1002/qj.3155, 2017.

Peings, Y., Z. M. Labe, and G. Magnusdottir, 2021: Are 100 Ensemble Members Enough to Capture the Remote Atmospheric Response to +2°C Arctic Sea Ice Loss? Journal of Climate, 34, 3751–3769, https://doi.org/10.1175/JCLI-D-20-0613.1.

Perez, J., Maycock, A. C., Griffiths, S. D., Hardiman, S. C., and McKenna, C. M. (2024). A new characterisation of the North Atlantic eddy-driven jet using two-dimensional moment analysis, Weather Clim. Dynam., 5, 1061–1078, https://doi.org/10.5194/wcd-5-1061-2024.

Saffin, L., C. M. McKenna, R. Bonnet, and A. C. Maycock, 2024: Large Uncertainties When Diagnosing the "Eddy Feedback Parameter" and Its Role in the Signal-To-Noise Paradox. Geophysical Research Letters, 51, e2024GL108861, https://doi.org/10.1029/2024GL108861.

Shaw, T. A.: Mechanisms of Future Predicted Changes in the Zonal Mean Mid-Latitude Circulation, Curr. Clim. Change Rep., 5, 345–357, https://doi.org/10.1007/s40641-019-00145-8, 2019.

Siew, P. Y. F., Li, C., Sobolowski, S. P., King, M. P. (2020). Intermittency of Arctic–mid-latitude teleconnections: stratospheric pathway between autumn sea ice and the winter North Atlantic Oscillation. Weather and Climate Dynamics, 1(1), 261-275. https://doi.org/10.5194/wcd-1-261-2020.

Ye, K., Woollings, T., and Screen, J. A. (2023). European Winter Climate Response to Projected Arctic Sea-Ice Loss Strongly Shaped by Change in the North Atlantic Jet, Geophys. Res. Lett., 50, e2022GL102005, https://doi.org/10.1029/2022GL102005

Ye, K., T. Woollings, S. N. Sparrow, P. A. G. Watson, and J. A. Screen, 2024: Response of winter climate and extreme weather to projected Arctic sea-ice loss in very large-ensemble climate model simulations. npj Clim Atmos Sci, 7, 1–16, https://doi.org/10.1038/s41612-023-00562-5.

---

## Author Response (AR2)

**Reviewer Response 'Minimal influence of future Arctic sea ice loss on North Atlantic jet stream morphology' submitted to Weather and Climate Dynamics**

Yvonne Anderson, Jacob Perez, Amanda C. Maycock

January 2025

We thank the Editor for taking the time to review the paper and for providing this additional feedback. Comments are in black and author responses are in blue.

**Editor Comments**

**Comment 1**

I believe there is an opportunity to enhance the manuscript by further exploring the mechanisms of sea ice impact on the wind speed response in Fig.2. Specifically, you mention in the text that the change in sea ice leads to alterations in the temperature gradient, but how it varies across different models is not shown. I suggest exploring how the atmospheric temperature and gradients in the lower troposphere (around 850 hPa) change in response to SIC changes in each of the six models. This would provide further insight into the differences in jet response, particularly why the response is much weaker in MIROC6 and AWI-CM-1-1-MR compared to other models. I am particularly interested in whether the weakening of wind speed on the poleward side of the jet is related to changes in the local temperature gradient around the Greenland coastline, or if it reflects an equatorward shift of the jet due to changes in the large-scale equator-to-pole gradient that alter the meridional circulation.

Figure R1 shows the atmospheric temperature response to sea ice loss in the lower troposphere. For each model we calculated the ensemble mean difference in 850 hPa air temperature between present and future simulations. All models show significant positive air temperature anomalies between 50-70 °N including AWI-CM-1-1-MR and MIROC6, which showed the weakest zonal wind response.
We define a low and high latitude temperature gradient to explore whether changes in temperature gradient with sea ice loss are related to the zonal wind speed response. The low latitude air temperature gradient is calculated by selecting air temperature anomalies in two boxes at 20-30°N and 40-50°N. We then take the mean across the North Atlantic region (0-60°W) or the whole hemisphere (0-360°) in both boxes and take the difference between the boxes. Likewise, the high latitude temperature gradient is calculated between boxes at 44-54°N and 64-74°N. Air temperature gradients were compared to the wind response at low and high latitudes. The latitude bands are chosen to correspond to the temperature gradient across the low (30-39°N) and high (54-63°N) latitude nodes of the zonal wind response index shown in Fig 2.

[Figure]

Figure R1: DJF ensemble mean air temperature anomalies at 850 hPa in PAMIP models between simulations forced by present-day and future SIC. Stippling indicates grid points where the difference is significant based on a two-sample Student's t-test at the 95% confidence level.

Figure R2 shows the low latitude temperature gradient plotted against the low latitude wind response to sea ice loss. In the North Atlantic region, the low latitude temperature anomaly decreases in strength from FGOALS-f3-L, MIROC6, AWI-CM-1-1-MR, IPSL-CM6A-LR, CanESM5, HadGEM3-GC31-MM. In the hemispheric zonal mean, the low latitude temperature anomaly decreases in strength from IPSL-CM6A-LR, AWI-CM-1-1-MR, CanESM5, FGOALS-f3-L, MIROC6 and HadGEM3-GC31-MM. There is no clear intermodel relationship between the strength of the low latitude temperature gradient and the low latitude wind response both for the North Atlantic region and the zonal mean.

[Figure]

Figure R2: Relationship between low latitude temperature gradient with sea ice loss and low latitude wind response at 850 hPa. Filled circles shows the response in the North Atlantic region, crosses show the response in the hemispheric zonal mean.

Figure R3 shows the relationship between the high latitude temperature gradient and the high latitude wind response. In the North Atlantic region, the high latitude temperature gradient is always negative, as expected, and decreases in strength from CanESM5, FGOALS-f3-L, MIROC6, IPSL-CM6A-LR, AWI-CM-1-1-MR and HadGEM3-GC31-MM. In the zonal mean, the high latitude temperature gradient is always negative and strongest from HadGEM3-GC31-MM, CanESM5, AWI-CM-1-1-MR, MIROC, IPSL-CM6A-LR and FGOALS-f3-L. There is no clear relationship between the strength of the high latitude temperature gradient and the high latitude wind response at 850 hPa.

From this analysis, the weaker zonal wind response in MIROC6 and AWI-CM-1-1-MR cannot be explained by differences in 850 hPa air temperature gradient. We then explored the relationship between the air temperature gradient at 850 hPa and wind anomalies at 500 and 700 hPa (Figures R4-7), to assess whether the wind shear is related to the temperature gradient. However, we see no clear relationship between the temperature gradient and wind response at these different levels. Given the inconclusive findings, we have not added this further analysis to the manuscript as we do not feel it strengthens the results.

[Figure]

Figure R3: Relationship between high latitude temperature gradient with sea ice loss and high latitude wind response at 850 hPa. Filled circles shows the response in the North Atlantic region, crosses show the response in the hemispheric zonal mean.

[Figure]

Figure R4: Relationship between low latitude temperature gradient at 850 hPa and low latitude wind response at 700 hPa. Filled circles shows the response in the North Atlantic region, crosses show the response in the hemispheric zonal mean.

[Figure]

Figure R5: Relationship between high latitude temperature gradient at 850 hPa and high latitude wind response at 700 hPa. Filled circles shows the response in the North Atlantic region, crosses show the response in the hemispheric zonal mean.

[Figure]

Figure R6: Relationship between low latitude temperature gradient at 850 hPa and low latitude wind response at 500 hPa. Filled circles shows the response in the North Atlantic region, crosses show the response in the hemispheric zonal mean.

[Figure]

Figure R7: Relationship between high latitude temperature gradient at 850 hPa and high latitude wind response at 500 hPa. Filled circles shows the response in the North Atlantic region, crosses show the response in the hemispheric zonal mean.

**Comment 2**

My other suggestion is minor, and it could simply reflect my limited understanding of the PAMIP experiments, but I think it would benefit the manuscript to provide a bit more description of the experimental design to avoid any confusion for readers. From my understanding, in PAMIP1.6, SSTs are set to present-day conditions, while SIC is set to reflect a future warming of 1.4°C relative to present-day climate. I am wondering, though, whether SIC is the same for all models, regardless of their climatology, or if the SIC anomalies are added to each model's baseline climatology, so the actual SIC field would differ across models. For those who are less familiar with the setup, it might be helpful to clarify this point further in the text. I did look at Smith et al. (2019), but I was still unsure, as they showed SIC anomalies in Fig. 5-6c,f, but the text implied that SIC forcing was the same for all models, regardless of their individual climates. Additionally, following this line of thought, it may be worth revising the legend in Figures 6-7 to change the label from 'future' to 'future SIC'. This would help clarify that these PAMIP1.6 experiments are forced by SIC changes only, and do not incorporate other climate changes (e.g., GHGs) if that is correct.

We thank the Editor for their suggestion of clarifying this section about the datasets and methods. Each model simulates their own present-day and future SIC fields, however the final SIC forcing applied in the experiments is constrained by observed present-day SIC values. Firstly, global mean temperatures for pre-industrial, present-day and future time periods are defined as 13.67°C, 14.24°C and 15.67°C, respectively. For each model, the 30 year period where the average global temperature matches these values is then determined. The average SIC during those periods is then calculated to give pre-industrial, present-day and future SIC fields. At each grid point, linear regression is calculated between simulated future and present-day SIC. The point where the regression line intersects the observed present-day value is taken as the future SIC estimate. The outcome is a single, consistent SIC forcing field applied to all models, rather than one that reflects the unique climatology of each model. This information can be found in Smith et al. (2019), Appendix A.

We have clarified this point in Section 2.1 of the manuscript as follows: "Taking the ensemble mean SIC for CMIP5 simulations results in a poor representation of the ice edge. To address this issue, future SIC projections are constrained by present-day observations to ensure a more accurate representation. For each model, linear regression is calculated between simulated future and present-day SIC at each grid point. The future SIC estimate is taken as the point where the regression line intersects with the present-day observed SIC value. The outcome is a single, consistent SIC forcing field applied to all models, rather than one that reflects the unique climatology of each model. Additionally, in regions where the difference between present-day and future SIC is greater than 10%, present-day SSTs are replaced by future SSTs in experiment 1.6."

To address your second point, the authors agree that revising the legend in Figure 6-7 to specify "future SIC" is helpful. This change will clarify that the PAMIP 1.6 experiments are forced by changes in SIC only.

**Final reviewer comments**

Thanks for clarifying the ZWRI index. I'd like to suggest a minor further revision to this sentence: "However, changes in zonal wind in other regions may not reflect the local North Atlantic eddy-driven jet response." I think something like this is even more clear (at least to me): "However, changes in the

hemispheric zonal winds may not reflect the local North Atlantic eddy-driven jet response". In the sentence just prior to this you could also add "hemispheric", so "based on the hemispheric zonal mean zonal wind". The point being that writing "zonal mean" with no comment does not automatically mean readers will infer that you are talking about all longitudes, as opposed to just in a particular region (which is why I was confused in the first place).
In L348 there's a word missing ("the to the").

The authors thank the reviewer for their additional feedback and agree that this clarification will be helpful. The manuscript has been updated to include 'hemispheric' to further clarify the definition of the zonal mean zonal wind.

**References**

Smith, D. M., Screen, J. A., Deser, C., Cohen, J., Fyfe, J. C., García-Serrano, J., Jung, T., Kattsov, V., Matei, D., Msadek, R., Peings, Y., Sigmond, M., Ukita, J., Yoon, J.-H., and Zhang, X.: The Polar Amplification Model Intercomparison Project (PAMIP) contribution to CMIP6: investigating the causes and consequences of polar amplification, Geosci. Model Dev., 12, 1139–1164, https://doi.org/10.5194/gmd-12-1139-2019, 2019.

---

## Author Response (AR3)

**Reviewer Response 'Minimal influence of future Arctic sea ice loss on North Atlantic jet stream morphology' submitted to Weather and Climate Dynamics**

Yvonne Anderson, Jacob Perez, Amanda C. Maycock

February 2025

We thank the Editor for taking the time to consider our replies and for providing this additional feedback. Comments are in black and author responses are in blue.

**Editor Comments**

Thank you for responding to my comments. I'd like to further clarify whether you tested the sensitivity of the wind response to the latitudes used to calculate the temperature gradient. Specifically, you used the difference between 20–30°N and 40–50°N. I assume these latitudes were chosen because the strongest wind increase occurred between 30-40°N, even though these latitudes were less impacted by the changes in sea ice conditions (as shown in Fig. R1). Did you also examine latitudes further north, particularly around 60°N, where the largest wind decrease on the poleward side of the jet is observed? Perhaps that can be better explained by the sea ice change.

Yes, we examined the wind response near 60°N using a high latitude temperature gradient calculated as the temperature difference between 44-54°N and 64-74°N. These regions were chosen to coincide with the high latitude node of the zonal wind response index used in Figure 2 of the manuscript. However, there is also no relationship between the high latitude temperature gradient and the wind response at 500, 700 and 850 hPa (Figure R1) suggesting remote influences must dominate the wind response in this region.

While, as you mentioned in the paper, the wind speed in the core of the jet showed minimal changes, I believe the weakening (strengthening) of wind speed on the poleward (equatorward) flank of the jet is important, particularly for regional climates. Given that it is a fairly robust signal across the models as can be seen in Fig. 3 and 4, it might be worth highlighting this more in the paper (e.g., by mentioning that in the abstract).

We agree that these changes in the zonal mean wind should be better highlighted in the manuscript. Following the Editor's suggestion, we have added the following sentence to the abstract:
"Four of six models show a significant weakening of the westerlies on the poleward side of the North Atlantic jet and a strengthening on the equatorward side. However, there is no change in jet speed

and jet tilt across all models and no robust change in jet mass (area-weighted speed) when using the feature-based jet identification."

[Figure]

Figure R1: Relationship between high latitude temperature gradient change and high latitude wind response at 850 hPa due to future sea ice loss. Dots show the response in the North Atlantic sector and crosses show the zonal mean. Neither set of points show a significant inter-model correlation.